# Evaluation of Urban Sustainability in Cities of The French Way of Saint James (Camino de Santiago Francés) in Castilla y León according to The Spanish Urban Agenda

Francisco Tomatis [1,*], Luisa F. Lozano-Castellanos [1], Oscar L. García-Navarrete [1,2], Adriana Correa-Guimaraes [1], Maria Sol Wilhelm [3], Ouiam Fatiha Boukharta [1], Diana A. Murcia Velasco [1,2] and José E. Méndez-Vanegas [4]

1   Department of Agricultural and Forestry Engineering, University of Valladolid, UVa, University Campus of Palencia, 34004 Palencia, Spain; luisafernanda.lozano@uva.es (L.F.L.-C.); oscarleonardo.garcia@uva.es (O.L.G.-N.); adriana.correa@uva.es (A.C.-G.); ouiamfatiha.boukharta@uva.es (O.F.B.); dianaalexandra.murcia@alumnos.uva.es (D.A.M.V.)
2   Departamento de Ingeniería Civil y Agrícola, Facultad de Ingeniería, Universidad Nacional de Colombia, Bogotá 111321, Colombia
3   Centro de Estudios de Variabilidad y Cambio Climático, Facultad de Ingeniería y Ciencias Hídricas, Universidad Nacional del Litoral, Santa Fe 3000, Argentina; msolwilhelm@gmail.com
4   Investigation Group INYUMACIZO, Natural Resources Administration Subdirectorate, Regional Autonomous Corporation of Tolima—CORTOLIMA, National Open and Distance University, Ibagué 730006, Colombia; evelio.mendez@cortolima.gov.co
*   Correspondence: francisco.tomatis@uva.es

**Abstract:** The emblematic French Way of Saint James (Camino de Santiago Francés) crosses towns, cities, and Spanish regions to the Cathedral of Santiago de Compostela (Galicia, Spain), However, *where is The French Way of Saint James going with respect to the urban sustainability of its host cities?* As each city is unique and urban sustainability favors the revitalization and transition of urban areas, to know where to go, it is first necessary to establish a diagnosis that makes the different urban situations visible. In this article, the behavior of urban sustainability is analyzed in the six host cities of The French Way of Saint James in the Autonomous Community of Castilla y León, a region characterized by its link with the rural environment and its current depopulation problems. The data and indicators used are officially provided by the Spanish Urban Agenda, which, through the normalization of its values, are able to territorialize the SDGs at the local level and reflect the realities of the cities of Burgos, Astorga, Cacabelos, León, Ponferrada, and Valverde de la Virgen. The results make it possible to diagnose and compare these host cities, identifying weaknesses, skills, and opportunities that favor the promotion of action plans, local or joint (favored by The French Way of Saint James), in the multiple aspects of sustainability. In addition, they show that Valverde de la Virgen is the city with the best performance in terms of urban sustainability.

**Keywords:** sustainability; urban sustainability; SDG; Spanish Urban Agenda; The Way of Saint James; Camino de Santiago

## 1. Introduction

The challenges of urbanization and demographic change are increasingly recognized as components of resilient and sustainable development [1], especially in a context where the world's population is expected to reach 10 billion people by the year 2050, with 68% located in urban areas [2]. The society and the world of the 21st century, which are eminently urban, are "incorporating one of the deepest and most accelerated transformations in the history of humanity" [3]. There is no doubt that we are facing a major epochal change [4].

It is urgent to change the current models of urban development from the perspective of sustainable development, climate change, and resilience to natural and technological

disasters, and improve the quality of life, social integration, and equity [5–7]. However, to know what we need to change, we need to know where we are and where we are going.

Today, around 75% of the European population lives in urban areas. Estimates predict that the European urban population will increase to 80% by 2050 [8]. For this, it is vital to ensure a sustainable urban environment, knowing that cities are the engines of Europe's economy and are increasingly recognized as key players in the transition to a low-carbon economy [8]. The EU has a key role in promoting sustainable urban development [8] and considers that urban environmental sustainability encourages the revitalization and transition of urban areas and cities to improve the quality of life, promote innovation, and reduce environmental impacts while it maximizes economic and social co-benefits. The European Environment Agency considers urban sustainability from an environmental perspective achieved by focusing on environmental issues in urban areas such as air and water pollution, green spaces providing space for people and nature, biodiversity loss, resource efficiency, and mitigation measures to reduce greenhouse gas emissions and manage the impacts of climate change (Figure 1).

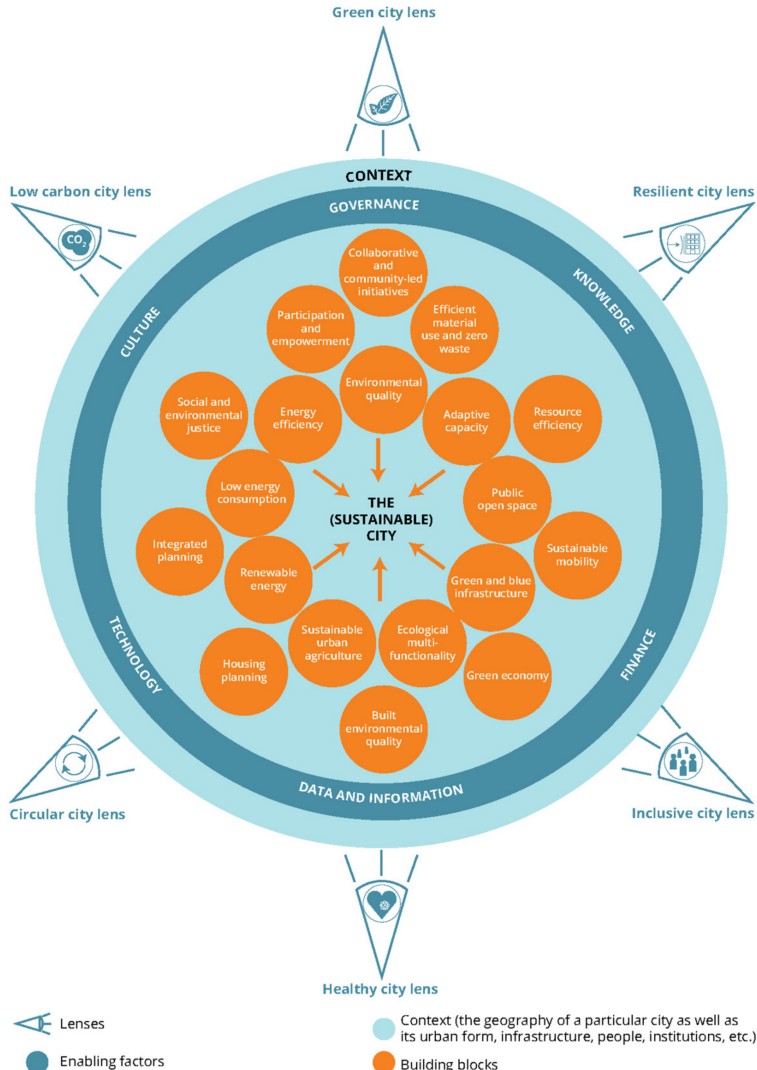

**Figure 1.** Conceptual framework for urban environmental sustainability. Source: European Environment Agency, 2021 [8].

In Spain, according to the Spanish Urban Agenda of 2019 [9], an important document in this article, the population percentages mentioned recently have already been reached. Spain is among the countries with the highest percentage of urban population in the

European Union (EU): of the 46,528,024 people that exist in the Spanish territory, 80% are concentrated in urban areas, which account for only 20% of the territory [9]. Spanish cities constitute, in many aspects, a reference in the European context due to their traditional compact city model, and a long tradition of urban planning and policy that has allowed rigorous action to be taken in the regeneration of centers and interior reform [10].

In this context, the search for sustainable development at the urban level and how cities, with their diverse conditions, characteristics, and available resources (sometimes limited), can move towards their own sustainability are of interest.

At the international level, the Sustainable Development Goals (SDGs) have emerged as a universal call to improve lives around the world. In 2015, all member states of the United Nations (UN) approved 17 SDGs, consisting of 169 targets and many indicators [11], as part of the 2030 Agenda for Sustainable Development. Since its launch, progress has been made on many SDGs. Nevertheless, the overall action to achieve all the goals is still not moving at the right speed and scale [12]. Even before the COVID-19 pandemic, which has unleashed an unprecedented social, environmental, and economic crisis, progress towards the SDGs was uneven and not on track to meet them by 2030.

Spain, similar to most European governments, is committed to the SDGs and the 2030 Agenda. In fact, the National Government of Spain has made an official document called the "Spanish Urban Agenda" (2019) that serves as a reference framework for urban areas with a focus on sustainable development and the 2030 Agenda. The Spanish Urban Agenda (SUA) stands out as a roadmap that can show the way to make Spanish towns and cities friendly, welcoming, healthy, and conscientious areas of coexistence by 2030. Although the SUA is focused on SDG 11 "*Make cities and human settlements inclusive, safe, resilient and sustainable*", it is also fully aligned, directly and indirectly, with the other SDGs.

The SUA is linked to the European Urban Agenda, which appears as an important reference due to its innovative methodology based on the principles of multilevel governance [10]. As a key objective, the adaptation of the economic programs of the European Union to the real challenges of the cities is useful in defining future actions that propose that the Spanish Urban Agenda adopts the necessary changes needed by the planning and management of urban policies [10]. The promotion by the European and Spanish institutions of a sustainable city model constitutes an opportunity to implement local Urban Agendas as documents that are required to structure an integrated vision of local urban policies from which different urban plans, mobility, energy, housing, environments, and economy can be developed [10].

Furthermore, the SUA is a very useful official tool for analyzing Spanish cities in terms of sustainability, as it has a system of indicators and a series of guidelines that allow diagnosis, monitoring, and revision of its own contents. On the one hand, the SUA presents purely descriptive data that is supplied by the General State Administration and, on the other hand, the SUA presents evaluation and monitoring data. This is important because urban areas are considered to be a reflection of the victories or defeats in the battle for sustainability [9] and its success will depend largely on the ability to reorient current urbanization processes from public management and policies [7]. This sentence outlines a challenge that can be faced with the economic tools and knowledge but whose success will ultimately depend on the collective political [7]. Considering the SUA as an excellent opportunity to incorporate sustainable development in Spanish cities, this article uses its data to analyze the host cities of The French Way of Saint James (Camino de Santiago Francés in the Spanish language) in the Autonomous Community of Castilla y León (according to the area affected by the declaration of the historic set of The French Way of Saint James in Castilla y León) [13]. From this perspective, the enhancement of cultural heritage can play a decisive role, not only in terms of increasing the life cycle of the heritage but also as an urban strategy capable of generating new economic, cultural, and social values, supporting the innovative dynamics of local development [14].

The SUA presents a territorial diagnosis and synthesis and descriptive indicators for cities with 5000 people or more (considering the urban population of 2019). Therefore,

the cities contemplated in this analysis are Astorga, Cacabelos, León, Ponferrada, and Valverde de la Virgen (from the Province of León) and the city of Burgos (from the Province of Burgos). It should be clarified that Cababelos, in 2019, had a population of more than 5000 people; however, in the last 2 years, its population has been less than 5000. The Province of Palencia does not have cities with more than 5000 people through which The French Way of Saint James passes, so it is exempt from this evaluation (Table 1).

**Table 1.** Cities with more than 5000 people in 2019 that are hosts of The French Way of Saint James in the Autonomous Community of Castilla y León (Spain).

| Province of León | Province of Palencia | Province of Burgos |
|---|---|---|
| Astorga | | Burgos |
| Cababelos | | |
| León | | |
| Ponferrada | | |
| Valverde de la Virgen | | |

With the tools provided by the SUA, it is possible to territorialize and localize the commitments and challenges that are promoted at the international level in terms of sustainable urban development, within the framework of the SDGs, to local areas. From the visibility of the diagnoses and results obtained, the analyzed cities are given a preponderant position as the main promoters of their own sustainable development individually or together (taking advantage of the existing link with The French Way of Saint James), starting from the identification of aspects that are in line (or not) with the contribution of urban sustainability, SDGs, and the Specific Objectives of the SUA.

Through the data provided by the SUA in the context of connection with the SDGs, the diagnosis and analysis of the cities allows the promotion of bottom-up actions. In addition, having a reflection of the status of cities in terms of urban sustainability is useful to coherently promote various development strategies and/or action plans that manage to contribute to greater current and future socio-environmental well-being.

## 2. The Host Cities of The French Way of Saint James (Camino de Santiago Francés) in Castilla y León

One of the peculiarities that Castilla y León presents is the importance that The Way of Saint James has for many cities and towns. The Way of Saint James is an element of identity that gives a sense of belonging to the people and acts as a link between cities and towns, making them part of the same territorial-cultural system. In addition, Castilla y León is one of the areas most affected by depopulation in southern Europe [15] due to its greater agrarian tradition and the smaller initial size of its population centers [16].

The Way of Saint James is much more than a simple way. Hundreds of routes over eleven centuries have guided the journey of millions of people towards a particular meeting place: Santiago de Compostela in Galicia, Spain.

Today, the motivations of the people who travel the Way of Saint James are very diverse. Although, historically, the religious character has been highlighted as the main focus, nowadays, the Jacobean route has become a valuable tourist attraction, which pilgrims decide to travel to visit, as a traveler-tourist in general, the monuments and historical complexes, enjoy contact with well-cared-for nature and adequate accommodation, and even search for handcrafted souvenirs to take away [17]. There are as many routes as there are pilgrims to reach the Cathedral of Santiago de Compostela (Figure 2). However, The French Way of Saint James (Camino de Santiago Francés) is the best known, busiest, and best equipped route, which translates as the route preferred by the vast majority of pilgrims [18].

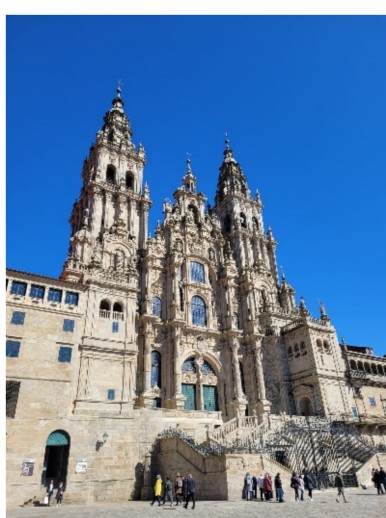

**Figure 2.** Cathedral of Santiago de Compostela in Galicia, Spain.

In Spain, from the Pyrenees (Roncesvalles) to Santiago de Compostela, The French Way has a length of approximately 750 km, which crosses five Spanish Autonomous Communities: Aragón, Navarra, La Rioja, Castilla y León, and Galicia [18]. Within the total distance covered by The French Way in Spain, approximately 400 km, more than half of the route, is in Castilla y León [19]. From east to west, the road passes through Castilla y León through three of its provinces: 112 km in the Burgos Province, 70 km in the Palencia Province, and 212 km in the León Province [19].

As part of the alternatives of the Jacobean route, it is noted that in Castilla y León, there are also other alternative ways: Silver Way (Camino Vía de la Plata), Portuguese Way (Camino Portugués), Mozárabe-Sanabrés Way (Camino Mozárabe-Sanabrés), Vía de Bayona Way (Camino Vía de Bayona), the Besaya Way (Camino del Besaya), Vadiniense Way (Camino Vadiniense), Salvador Way (Camino del Salvador), Madrid Way (Camino de Madrid), Levante and Sureste Way (Camino de Levante y Sureste), La Lana Way (Camino de La Lana), and Real de Invierno Way (Camino Real de Invierno).

The Way of Saint James in Castilla y León represents one of the greatest surprises offered to pilgrims, travelers, and tourists. The route stretches through fields, rises through mountains, and flows through rivers, which makes it an outstanding place due to its landscape and natural values. Today, while the natural route that is The Way of Saint James has rising value, the path, in its passage through this autonomous community, is a varied reflection of counties and a multicolored explosion of different spaces that allow an exceptional route [20]. In addition to its natural values, the Jacobean route in Castilla y León has a very rich artistic heritage, where the predominant style is Romanesque [20].

The Way of Saint James is very important for the survival and development of the cities and towns of Castilla y León. However, the reality of the provinces, cities, and towns of Castilla y León currently presents the social challenges of depopulation and aging [21]. This is a localized reality in Castilla y León because at the national level, in Spain, the population has tended to increase in recent years. Actually, politically and socially, it is even called "Castilla y León empty and/or emptied" [22].

Castilla y León is extremely rural: 2115 municipalities (94% of the total) have a population of less than 2000 people [15]. In total, 79.8% of the municipalities in Castilla y León do not exceed 100 people. Only 9 municipalities exceed 50,001 people, of which only 4 exceed 100,000 people [22]. Thus, depopulation has become a real problem of the state, which requires the implementation of innovative public policies aimed at boosting the local economy, providing basic and quality social services to rural areas [23].

Faced with this situation of rurality, intermediate cities acquire importance, considering that the concept of an "intermediate city" transcends the scope of spatial and population size to open new perspectives that modify the hierarchy of scalar analysis. This concept

also includes the analysis of the economic, social, cultural, and environmental governance of cities that, due to their territorial implantation, have to play a key role in correcting inequalities derived from current urbanization, which is too focused on large cities [24]. The most holistic definition [25,26] is summarized below: (a) The intermediate city, beyond its demographic relevance, has the capacity to structure and unite the urban system and urban–rural links. Intermediate cities weave and work in networks. (b) The intermediate city, due to its scale, has a greater capacity to draw up and implement high-benefit strategies that allow it to position itself in regional, national, and even international scenarios, using fewer resources than large cities. (c) The intermediate city constitutes in itself a groundbreaking element of the status quo derived from the impact of globalization, since it contributes to questioning the hierarchies of the urban system, opening new horizons of territorial cooperation [24,27].

If we consider the host analyzed cities of Burgos, León, Ponferrada, Astorga, Valverde de la Virgen, and Cacabelos as intermediate cities, then they are cities with a remarkable potential for dynamism, which could experience growth in their population and activities [28]. However, when analyzing the population, Leon and Astorga have shown depopulation in recent years (Figure 3) while Ponferrada, Valverde de la Virgen, Burgos, and Cacabelos have shown population increases (Figure 4).

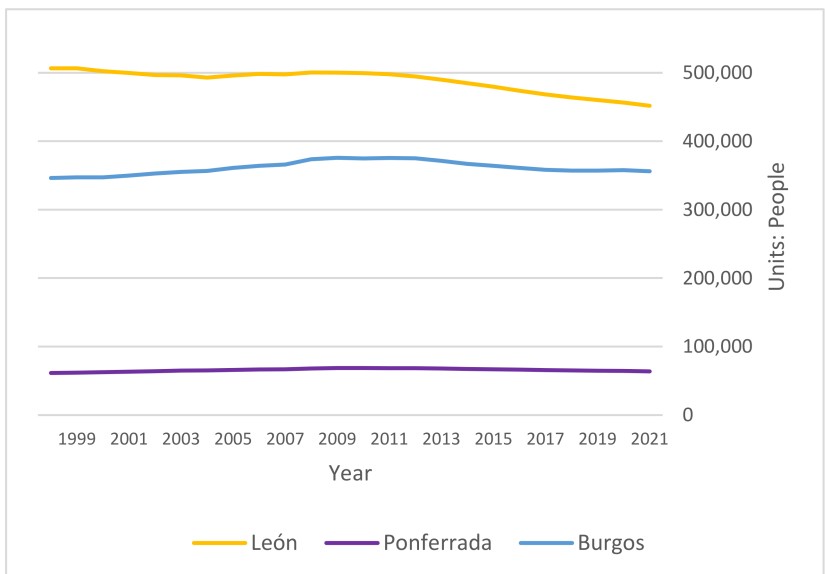

**Figure 3.** Population evolution in the cities of León, Burgos, and Ponferrada (1998–2021). Source: Own elaboration with data extracted from the INE website: www.ine.es (accessed on 15 July 2022) [21].

Taking into account that the cities analyzed share the passage of The Way of Saint James (host cities) and have more than 5000 people living there (at least for the year 2019), they have diverse characteristics:

- Burgos and León are provincial capitals and represent the upper segment of the urban centers of the Autonomous Community of Castilla y León [29]. They have more than 300,000 people.
- Ponferrada had 63,747 people in the year 2021 [30] and articulates large territorial areas. Together with Burgos and León, they are intermediate urban areas located in the northwest of the peninsula [28].
- Astorga, Cacabelos, and Valverde de la Virgen have fewer than 15,000 people and represent the smallest cities in this analysis. They have importance and transcendence within the province of León.

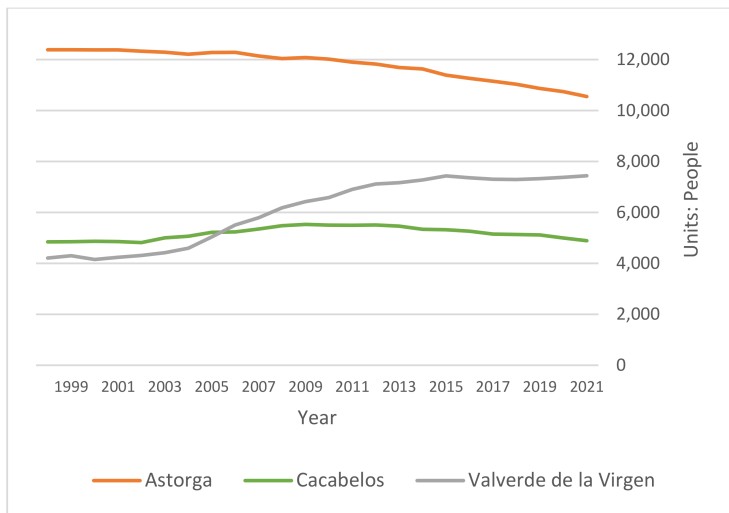

**Figure 4.** Population evolution in the cities of Astorga, Cacabelos, and Valverde de la Virgen (1998–2021). Source: Own elaboration with data extracted from the INE website: www.ine.es (accessed on 15 July 2022) [21].

In Spain, the presence of medium/intermediate cities is the real constitutive element of the urban network and one of the most important explanatory bases of the territorial structure of the country and they are the territorial labels of the "glocal" [28]. Considering that the cities mentioned are important for the development of the region, the search for urban sustainability, aligned with the cultural-territorial union of The Way of Saint James, is an alternative to promote joint territorial actions, action plans, and even receive EU aid that can bring economic, social, and environmental benefits for their future development.

## 3. The Challenge of the SDGs and the Implementation of Indicators at the Local Level

The literature on indicators in general, and the literature on indicators in urban contexts shows that sustainability indicators are used to monitor change in society and show progress towards a given goal or objective, based on observable or measurable markers [31–34]. Indeed, an indicator can be defined as an observable characteristic that is supposed to represent a generally unobservable state or trend at a given time [35,36].

In particular, the SDG indicators pose great challenges regarding their implementation at the local level [37]. Although they recognize the role of cities in sustainable development, the 2030 Agenda was agreed and signed by national governments. So, the territorialized implementation of its targets and indicators requires a process of adaptation and/or localization [38].

The challenges mentioned specifically refer to the availability and access of reliable, spatial, specific, standardized, open, comparable, measurable, and observable data at the urban level [38–44]. This is often framed by a lack of solid institutions dedicated to data collection at the city scale [44]. Therefore, the challenges of accessing the data located at the local level must include the selection of a set of limited indicators that manage to reflect aspects necessary to evaluate the sustainability of a city [42]. In addition, the data must necessarily be simplified for monitoring so that it can be replicated in several cities [43], which would allow cities to be compared based on the relevant evaluations.

In this context, SUA appears to be a very useful tool in this field since it transfers the urban goals of the SDGs of the 2030 Agenda, especially SDG 11, to the reality of towns and cities. The 17 SDGs are:

1.  No poverty
2.  Zero hunger
3.  Good health and well-being
4.  Quality education
5.  Gender equality

6.  Clean water and sanitation
7.  Affordable and clean energy
8.  Decent work and economic growth
9.  Industry, innovation and infrastructure
10. Reduced inequalities
11. Sustainable cities and communities
12. Responsible consumption and production
13. Climate Action
14. Life below water
15. Life on land
16. Peace, justice and strong institutions
17. Partnerships for the goals.

The SUA even has its own objectives and indicators that are aligned with the goals of the 2030 Agenda, making it a practical tool for achieving the SDGs at the local level. As shown in Table 2, the SUA links each of its 10 Specific Objectives to the SDGs [45].

**Table 2.** Relationship of each Specific Objective of the SUA with the SDGs.

| | Specific Objective SUA | Linked SDGs |
|---|---|---|
| 1. | Organize the territory and make rational use of the land, conserve and protect it | 2; 6; 11; 14; 15 |
| 2. | Avoid urban sprawl and revitalize the existing city | 1; 4; 11 |
| 3. | Prevent and reduce the impacts of climate change and improve resilience | 1; 3; 11; 13 |
| 4. | Make sustainable management of resources and favor the circular economy | 6; 7; 11; 12 |
| 5. | Promote proximity and sustainable mobility | 3; 9; 11 |
| 6. | Promote social cohesion and seek equity | 1; 4; 5; 10; 11 |
| 7. | Promote and favor the Urban Economy | 1; 2; 5; 8; 9; 12 |
| 8. | Guarantee access to Housing | 11 |
| 9. | Lead and foster digital innovation | 5; 9 |
| 10. | Improve intervention instruments and governance | 4; 11; 12; 16; 17 |

Using data from an official government source such as the SUA and its consequent reflection of urban realities is what this research carries out. As increasingly more guidance is available on the localization of the SDGs [42,46,47] and research that highlights the challenges in implementing the SDGs at the local level, it is vital and necessary to acquire experiences to be able to work with the SDGs in practice and make them a tool to achieve sustainable development [42].

This article reflects the situations of the evaluated cities in achieving sustainable development and the Specific Objectives of the SUA, thus the SDGs, are identified. Based on the results obtained, cities have the capacity to influence local and regional conditions through policy intervention [48] in planning and decision-making, in creating awareness, in the encouragement of behavior, in the promotion of public participation [44], and in alignment with the SDGs. In any case, the realities of urban sustainability in a particular region that needs to become strong, unite, and work together for current and future problems and challenges are discovered and made visible.

## 4. Materials and Methods

### 4.1. Data Selection

Given the need to count and select quantitative, locally applicable, statistically comparable indicators from official sources, the SUA Descriptive Data was selected. They are related to the ten Specific Objectives that the SUA has.

Access to the SUA Database is free on the Spanish Urban Agenda website [3]. The Database has a table that collects information available from all the Spanish municipalities,

with more than 50 descriptive data on the SUA, and another table in which the data corresponding to the first quartile, the median value, and the third quartile are offered according to the cluster by population size: municipalities with more than 100,000 people, municipalities with between 50,000 and 100,000, between 20,000 and 50,000, between 5000 and 20,000 people, and municipalities with less than 5000 people. This database of over 200,000 data allows not only for data related to a one municipality to be obtained but also to make a comparison with municipalities with similar characteristics, such as the population.

Regarding the sources of information used to calculate the SUA Descriptive Database, the most recent data possible was used. These data can be requested from the Ministry of Transport, Mobility and Urban Agenda of the Government of Spain, and obtaining them facilitates an approximation to the current situation of each Spanish cities, configuring itself as a useful tool for decision-making and the establishment of specific goals to be achieved.

For the realization of this article, values were extracted (in July 2021) from the SUA Database and those that correspond to the cities of Castilla y León related to The French Way of Saint James were selected: Astorga, Cacabelos, León, Ponferrada, and Valverde de la Virgen (province of León) and the city of Burgos (province of Burgos). Of the 72 descriptive data made possible by the SUA Database, a total of 49 were considered for the analysis (Table 3). The rest of data was not available or did not indicate quantitative data to be analyzed, as is the case, for example, of the indicators D.38 and D.39 ("Date of the current urban planning figure in the municipality" and "Agenda Planning, Strategic Planning and Smart Cities", respectively).

Each of the SUA databases used, with its definition, relevance, and calculation methodology according to the SUA, are described in the document "Descriptive Data of the Spanish Urban Agenda" available on the SUA's web platform.

### 4.2. Data Normalization

The analysis carried out was located in places that, in addition to their geographical proximity, belong to the same Autonomous Community. Its relationship with The French Way of Saint James and its main distinctive characteristics are described in Section 2.

To make the cities comparable with the data and indicators provided by the SUA, they were reduced to a predominant non-dimensional unit for which the normalization method was used through the minimum-maximum index. Once the descriptive data of the SUA was collected, first, and following the steps considered by Nagy et al., 2018 [49], each of the variables was normalized on a scale of 0 to 10, where 0 indicates the worst performance and 10 the best performance. To eliminate the effect of extreme values, the lower and upper limits of each indicator were identified in each of the cities analyzed, and then the minimum-maximum method [50] was used, which makes it possible to create a range from 0 to 10.

Considering the main criteria of the New Urban Agenda [51], for some indicators (such as indicator D.05. "Green areas per 1000 people"), the high value score was considered to represent a good performance in terms of sustainability, marked as $\hat{x}$ in Table 3, and for others, it was considered that it represented a poor performance (such as indicator D.02.a. "Artificial coverage area by municipality"), marked as $\check{x}$ in Table 3. Therefore, the formulas were applied inversely depending on the attributes and thus also ensures that higher values represent better performance:

$$\hat{x} = \left( \frac{x - min(x)}{max(x) - min(x)} \right) * 10$$

$$\check{x} = \left( \frac{max(x) - x}{max(x) - min(x)} \right) * 10$$

where *x* is the raw data value; *min(x)* and *max(x)* determine the lower and upper limits for the worst and best performance, respectively; and $\hat{x}$ / $\check{x}$ is the normalized value after the rescaling process.

**Table 3.** Descriptive data of the SUA considered for the analysis.

| SUA Indicator | Source | Specific Objectives SUA | Max. Value | Min. Value |
|---|---|---|---|---|
| D.01. Population variation 2007–2017 (%) | INE | 1; 2; 3; 4; 5; 6; 7; 8; 9; 10 | 12.1 ($\hat{x}$) | −10.6 |
| D.02.a. Artificial coverage area by municipality (%). | CORINE | 1; 3 | 52.4 | 4.9 ($\check{x}$) |
| D.02.b. Crop area by municipality (%). | SIOSE | 1; 3 | 45.8 ($\hat{x}$) | 7.8 |
| D.02.d. Forest and meadows areas by municipality (%). | SIOSE | 1; 3 | 78.3 ($\hat{x}$) | 18.0 |
| D.03.a. Municipal surface destined to agricultural and forest exploitations (%). | SIOSE | 1; 3 | 0.4 ($\hat{x}$) | 0.0 |
| D.03.b. Surface destined to agricultural and forest exploitations with respect to the urban land and urbanizable delimited of the city (%). | SIOSE/SIU | 1; 3 | 5.6 ($\hat{x}$) | 0.0 |
| D.04. Municipal area of undeveloped land (%). | SIU | 1; 10 | 93.6 ($\hat{x}$) | 27.6 |
| D.05. Green areas per 1000 people. | SIOSE/INE | 1; 3 | 20.4 ($\hat{x}$) | 2.1 |
| D.06. Urban density. Number of people per hectare of urban land surface (people/ha) | INE/SIU | 1; 2; 4; 5; 6; 7; 8; 9 | 85.3 ($\hat{x}$) | 23.3 |
| D.07. Area of discontinuous mixed urban land over total mixed urban land (%) | SIOSE 2009 | 1; 2; 5 | 47.4 | 6.2 ($\check{x}$) |
| D.08. Housing density by urban land area (Hou./ha). | INE_CENSO/SIU | 1; 2; 4; 5; 6; 7; 8; 9 | 54.2 ($\hat{x}$) | 14.5 |
| D.09. Urban compactness. Total built area per floor area ($m^2t/m^2s$) | Catastro/SIU | 2; 5; 6 | 1.0 | 0.4 ($\check{x}$) |
| D.10.a. Constructed area for residential use by land area ($m^2t/m^2s$) | Catastro/SIU | 2; 5; 6 | 0.6 | 0.3 ($\check{x}$) |
| D.10.b. Built area for residential use with respect to the total built area (%). | Catastro/SIU | 2; 5; 6 | 75.2 ($\hat{x}$) | 43.3 |
| D.ST.01. Expected housing in development land areas (Hou./ha). | SIU | 2; 5; 6; 8 | 63.3 ($\hat{x}$) | 17.8 |
| D.ST.02. Percentage of development land areas with respect to the total urban land (%) | SIU | 1; 2; 10 | 94.3 ($\hat{x}$) | 2.0 |
| D.ST.03. Developable land delimited with respect to the total urban land (%) | SIU | 1; 2; 10 | 66.0 ($\hat{x}$) | 2.0 |
| D.ST.04. Percentage of land areas under development for residential use with respect to the total urban land (%). | SIU | 1; 2 | 66.5 ($\hat{x}$) | 1.5 |
| D.ST.05. Percentage of land areas under development used for economic activities (industrial or tertiary) with respect to the total urban land (%). | SIU | 1; 2; 6; 7 | 27.8 ($\hat{x}$) | 0.0 |
| D.14. Percentage of the building stock by municipality that was older than the year 2000 (%). | Catastro | 2; 3; 4 | 71.5 | 36.6 ($\check{x}$) |
| D.17.a. Transport infrastructure area (ha). | SIOSE | 1; 5 | 538.1 ($\hat{x}$) | 1.1 |
| D.17.b. Percentage of surface of transport infrastructures with respect to the municipal term (%) | SIOSE | 1; 5 | 5.4 ($\hat{x}$) | 0.0 |
| D.18.a. Residential vehicles per 1000 people. | DGT | 3; 5 | 621.1 | 506.0 ($\check{x}$) |
| D.18.b. Percentage of passenger cars (%) | DGT | 3; 5 | 78.2 | 70.1 ($\check{x}$) |
| D.18.c. Percentage of motorcycles (%) | DGT | 3; 5 | 8.2 | 6.7 ($\check{x}$) |
| D.22.a. Population aging index (%) | INE | 2; 5; 6; 7; 8; 9; 10 | 25.4 | 14.3 ($\check{x}$) |
| D.22.b. Population senescence index (%) | INE | 2; 5; 6; 7; 8; 9; 10 | 18.3 | 12.0 ($\check{x}$) |
| D.23. Percentage of foreign population (%) | INE | 2; 6; 7 | 7.5 ($\hat{x}$) | 3.0 |
| D.24.a. Total Dependency Index (%) | INE | 2; 6; 7 | 61.3 | 43.8 ($\check{x}$) |
| D.24.b. Child dependency ratio (%) | INE | 2; 6; 7 | 23.0 | 18.4 ($\check{x}$) |

**Table 3.** *Cont.*

| SUA Indicator | Source | Specific Objectives SUA | Max. Value | Min. Value |
|---|---|---|---|---|
| D.24.c. Elderly dependency ratio (%) | INE | 2; 6; 7 | 47.3 | 23.2 ($\check{x}$) |
| D.26.a. Workers in the agricultural sector (%). | INE | 6; 7 | 7.8 ($\hat{x}$) | 0.9 |
| D.26.b. Workers in the industrial sector (%). | INE | 6; 7 | 20.1 ($\hat{x}$) | 5.7 |
| D.26.c. Workers in the construction sector (%). | INE | 6; 7 | 10.5 ($\hat{x}$) | 4.5 |
| D.26.d. Workers in the service sector (%). | INE | 6; 7 | 89.0 ($\hat{x}$) | 68.1 |
| D.27.a. Establishments in the agricultural sector (%). | INE | 7; 9 | 8.2 ($\hat{x}$) | 0.1 |
| D.27.b. Establishments in the industrial sector (%). | INE | 7; 9 | 13.0 ($\hat{x}$) | 3.2 |
| D.27.c. Establishments in the construction sector (%). | INE | 7; 9 | 11.0 ($\hat{x}$) | 5.3 |
| D.27.d. Establishments in the service sector (%). | INE | 7; 9 | 91.3 ($\hat{x}$) | 71.2 |
| D.28.a. Total percentage of unemployed (%). | INE | 6; 7 | 16.0 | 10.0 ($\check{x}$) |
| D.28.b. Percentage of unemployed between 25–44 years (%) | INE | 6; 7 | 46.7 | 37.4 ($\check{x}$) |
| D.28.c. Female unemployment rate (%) | INE | 6; 7 | 58.4 | 55.7 ($\check{x}$) |
| D.29. Number of households per 1000 people. | INE | 2; 8 | 677.7 ($\hat{x}$) | 522.3 |
| D.32. Variation in the number of households 2001–2011 (%) | INE_Censo | 1; 2; 8 | 91.5 ($\hat{x}$) | 8.9 |
| D.33. Growth of the housing stock 2001–2011 (%) | INE_Censo | 1; 2; 4; 8 | 85.0 ($\hat{x}$) | 12.9 |
| D.34. Percentage of secondary households (%). | INE_Censo | 2; 8 | 13.9 ($\hat{x}$) | 6.2 |
| D.35. Percentage of empty housing (%). | INE_Censo | 2; 8 | 23.8 | 15.2 ($\check{x}$) |
| D.ST.06. Percentage of homes planned in development areas with respect to the existing housing stock (%) | SIU/INE_Censo | 1; 2; 4; 8 | 66.4 ($\hat{x}$) | 2.2 |
| D.ST.07. Number of homes planned in the development areas per 1000 people. | SIU/INE | 1; 2; 4; 8 | 386.2 ($\hat{x}$) | 14.9 |

Through normalization, the data became easily comparable between all the indicators. Therefore, values were obtained for each of the cities analyzed according to each of the 10 Specific Objectives of the SUA (mentioned in the Table 2). Using the arithmetic mean method using the values described, a general value per city was obtained and normalized on a scale of 0 to 10 according to its particular contribution to the SUA Objectives and, therefore, according to urban sustainability.

In Table 3, the indicators, the source, and the relationship with the Specific Objectives of the SUA are displayed. In addition, the maximum and minimum values of the group of cities analyzed without their respective normalization are revealed, in such a way that they serve as a reference framework to consider the ranges of real values that are handled between the cities of Astorga, Cacabelos, León, Ponferrada, Valverde de la Virgen, and Burgos.

## 5. Results

This research analysis focused on the urban sustainability of the host cities of The French Way of Saint James in Castilla y León, which in 2019 had more than 5000 people. The data provided by the SUA and each of the selected indicators (Table 3), considering their scores with the values already normalized and weighted for each city in the range of 0–10, allow us to identify and diagnose the situation of the cities according to the 10 objectives specific to the SUA and, therefore, to the SDGs (Table 2).

In this way, by analyzing the scores of each one of the indicators by the host city, grouped in the 10 Specific Objectives of the SUA, general values were obtained (Table 4).

According to the general values obtained, the city of Valverde de la Virgen has the best performance in terms of sustainability. Valverde de la Virgen also presents the best values in 9 of the 10 Specific Objectives of the SUA. In descending order, are the cities of León and Burgos, which present similar general values between them. Then, there is the city of

Ponferrada, which is the predecessor of the worst performers found in terms of general urban sustainability, visible in the cities of Cacabelos and Astorga, respectively (Figure 5).

**Table 4.** Results of the values referring to urban sustainability in the host cities, which were analyzed according to the Specific Objectives of the SUA.

| City | Specific Objectives of the SUA | | | | | | | | | | General Value |
|---|---|---|---|---|---|---|---|---|---|---|---|
| | 1 | 2 | 3 | 4 | 5 | 6 | 7 | 8 | 9 | 10 | |
| Burgos | 4.3 | 4.4 | 3.8 | 3.1 | 6.4 | 4.6 | 4.7 | 4.0 | 4.3 | 5.7 | **4.6** |
| Astorga | 3.8 | 2.9 | 4.6 | 0.7 | 3.3 | 3.6 | 3.3 | 2.4 | 2.9 | 2.7 | **3.0** |
| Cacabelos | 2.2 | 2.7 | 4.5 | 1.2 | 4.1 | 4.4 | 3.5 | 1.9 | 4.1 | 2.3 | **3.1** |
| León | 5.3 | 5.4 | 1.7 | 5.7 | 5.1 | 4.5 | 4.5 | 5.3 | 3.7 | 6.0 | **4.7** |
| Ponferrada | 3.4 | 4.3 | 3.9 | 3.0 | 4.3 | 4.7 | 4.1 | 4.0 | 3.5 | 4.1 | **3.9** |
| Valverde de la Virgen | 6.6 | 6.3 | 5.1 | 7.4 | 5.0 | 4.7 | 4.7 | 6.3 | 5.3 | 7.1 | **5.9** |

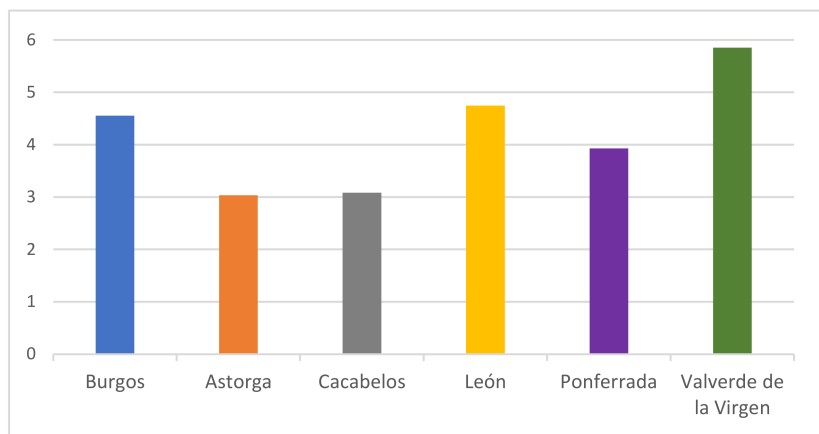

**Figure 5.** General values of the urban sustainability in the host cities of The French Way of Saint James in Castilla y León.

In detail, for each of the cities analyzed, Valverde de la Virgen has the best performance in all the Objectives (1; 2; 3; 4; 6; 7; 8; 9; and 10) compared to the other cities, with the sole exception of Objective 5. Its particular performance stands out in Specific Objective 4 "Make sustainable management of resources and favor the circular economy" and in Objective 10 "Improve intervention instruments and governance". In Specific Objectives 6 "Promote social cohesion and seek equity" and 7 "Promote and favor the Urban Economy", it is also a leader, but in this case, it shares its leadership with Ponferrada and Burgos, respectively. Continuing with the comparison, it is identified that the only Objective that the city of Valverde de la Virgen is surpassed is in Objective 5 "Promote proximity and sustainable mobility", where the cities of Burgos and León have the best diagnoses (Figure 6). Even in Objective 5, Valverde de la Virgen presented the lowest values and, therefore, the worst performance.

Following the leadership of Valverde de la Virgen, we visualized a group of cities composed of León and Burgos, which reflects their general values in terms of sustainability in a fairly similar way.

León is surpassed by Valverde de la Virgen in the Specific Objective 1 "Organize the territory and make rational use of the land, conserve and protect it", Objective 2 "Avoid urban sprawl and revitalize the existing city", Objective 4 "Make sustainable management of resources and promote the circular economy", and Objective 8 "Guarantee access to Housing". It is only surpassed by Burgos in Specific Objective 5, referring to "Promote proximity and sustainable mobility". However, there are no cities that present lower values than León in Specific Objective 3 regarding adaptation and resilience to climate change; therefore, the less favorable position in this particular aspect is attributed.

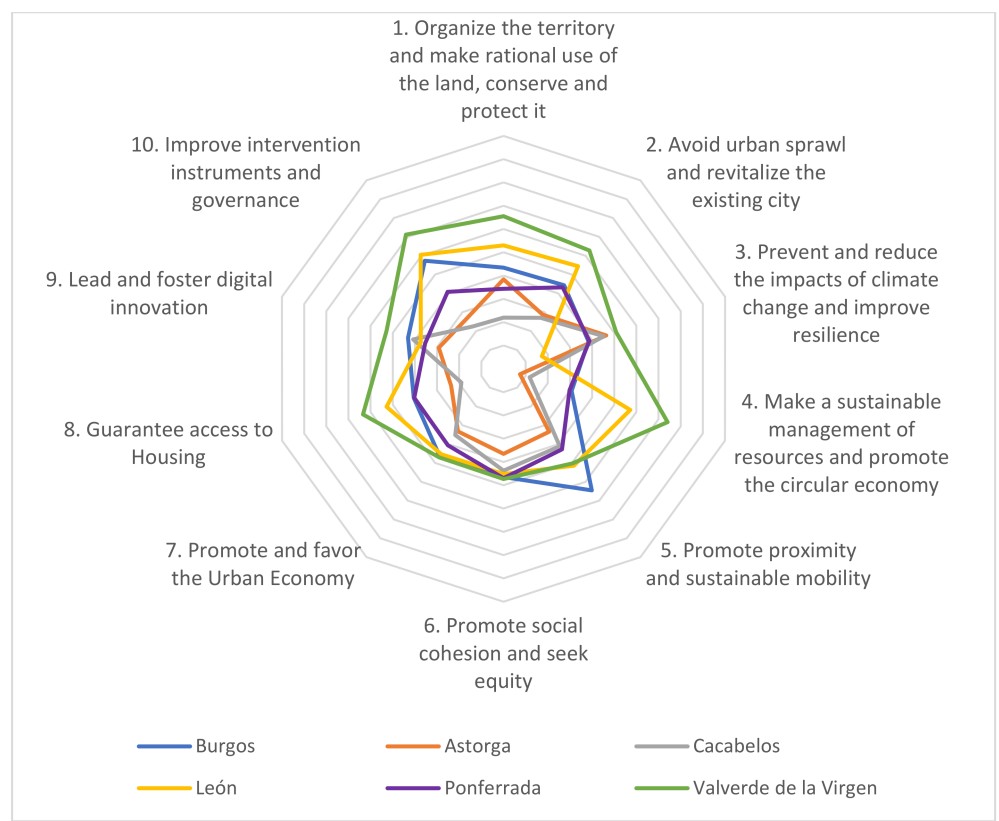

**Figure 6.** Radio visibility of the contribution to urban sustainability by city according to the Specific Objectives of the SUA.

Burgos, on the other hand, stands out as being the best city in the Specific Objective 5 regarding sustainable mobility and, together with Valverde de la Virgen, they present the best values in Objective 7 regarding urban economy. Without taking into account the comparison with the other cities, the values of Burgos appear quite uniform according to all the Specific Objectives of the SUA, where its best version is found in Objective 5 and its most unfavorable value in Objective 4 "Make sustainable management of resources and favor the circular economy".

Comparing the city of Ponferrada with the other cities, it is possible to observe that its position in fourth place, according to the general value obtained, represents and justifies its detailed analysis and comparison for each of the Specific Objectives. In fact, Ponferrada is in fourth position in Specific Objectives 2, 3, 4, 5, 7, 8 (with Burgos), 9, and 10. Undoubtedly, the particularity and the aspect to highlight of Ponferrada is there is no other city that surpasses it in its contribution towards Specific Objective 6, referring to "Promote social cohesion and seek equity". Viewing only the values relevant to Ponferrada, it is observed that its worst performance is shown under Objective 4 "Make sustainable management of resources and favor the circular economy".

The cities of Cacabelos and Astorga are the cities with the worst numbers in terms of the general values according to their contribution to the Specific Objectives defined by the SUA.

Compared to the other cities, Cacabelos presents the lowest values in compliance with Specific Objectives 1, 2, 8, and 10. In addition, it is in the penultimate place in its contribution towards Specific Objectives 4, 5, 6, and 7. Leaving aside the comparison, the worst value of Cacabelos is framed under Specific Objective 4, referring to "Make sustainable management of resources and promote the circular economy", and its best version is found in Specific Objective 3, regarding prevention, anticipation, and adaptation strategies to climate change.

For its part, the city of Astorga justifies its last position in terms of urban sustainability in this comparison of cities, since it shows the worst values in Specific Objectives 4, 5, 6, 7, and 9 and second to last in the values of Objectives 1, 2, 8, and 10. The city of Astorga is in second position, just behind Valverde de la Virgen, referring to its adaptive management of climate change, without a doubt the greatest aspect that contributes to its own urban sustainability. On the contrary, its diagnosis regarding the sustainable management of resources and the promotion of the circular economy shows the lowest value of all those presented by the other cities.

## 6. Discussion and Conclusions

This article used the data provided by the SUA to overcome the barriers and challenges that are usually found when trying to locate and territorialize the SDGs at the local level. The availability of reliable, quantifiable, and updated data, which serve as indicators, makes it possible to work with the SUA itself, giving it greater visibility and significance, in addition to stimulating progress "from the bottom up", considering cities as fundamental engines of change in terms of sustainability and the labels of the "glocal".

Understanding that the SDGs are more general and global objectives, their location and territorialization was carried out in this case by evaluating the Specific Objectives of the SUA, a document directly related to the 2030 Agenda due to its origin, nature, and objectives pursued. This work highlights the ability of the SUA to reflect the urban realities of Spanish towns and cities and to reflect their contributions towards sustainable development and the SDGs. In this case, host cities of Castilla y León, linked to The French Way of Saint James, were analyzed.

The results of the diagnosis of the cities of Valverde de la Virgen, León, Burgos, Ponferrada, and Cacabelos allowed the identification of four marked groups in reference to the state of the situation of sustainable development.

In the evaluation conducted, it is highlighted that the city of Valverde de la Virgen has the conditions and the support to lead other cities in terms of urban sustainability, a fact that perhaps its own authorities are unaware of. Especially, Valverde de la Virgen could lead and serve as an example of urban sustainability for small cities in the region such as Cacabelos and Astorga. Therefore, the analysis carried out thus gives it a very prominent role in the post of urban sustainability and even the city itself could cling to it, using it as a communication strategy to, for example, promote the passage of tourists who travel The French Way of Saint James through a "sustainable city". The good condition that Valverde de la Virgen presents in terms of sustainability is a great strength, considering the increase in population that it has experienced in recent years. In reference to this, it could improve its behavior in mobility and digital innovation since these are essential issues for a good-quality urban life in these times.

León and Burgos, beyond their geographical distance and belonging to different provinces, reflect a good performance in terms of sustainability with similar contributions towards sustainable development. These cities, given that both are capitals of their respective provinces and have more than 350,000 people, could also even be provincial, Autonomous Community, and The Way of Saint James benchmarks in terms of sustainability, and even work together to guide sustainable development at the regional level. In a potential joint effort for their sustainable development, León could learn from Burgos issues on sustainable mobility and climate change while Burgos could learn from León issues on resource management, circular economy, and access to housing such as urban land use. Since they are cities with a lot of weight in the development of the Autonomous Community, together, they have great potential to request aid for sustainability (for example, for climate change issues) from the Autonomous Community itself, the Government of Spain, and even the EU.

Ponferrada stands out as a city with uniform parameters. However, it could increase its efforts and ambitions to achieve a better performance. It is recommended that it aligns and works together with the cities that stand out the most in this regard, mainly with

León and Burgos due to their population sizes. With similar parameters considering the contributions towards the Specific Objectives of the SUA, the local administration presents the opportunity to define, based on the diagnosis, where to particularly focus the greatest efforts and action plans. However, it is recommended to promote actions in innovation, resource management, and the circular economy.

Both Cacabelos and Astorga are disadvantaged by the analysis carried out. Therefore, they are the cities that must begin to work more decisively in the search for urban sustainability so that they can guarantee a greater state of social, economic, and environmental well-being for their citizens, tourists, and biodiversity in general. The most disadvantaged position of these cities can also be taken to attract financing that motivates the implementation of ambitious and urgent action plans. It is recommended that they follow the exemplary performances of Valverde de la Virgen to identify and improve their good performances, replicate them, and work together.

Considering the general analysis of the cities, only Valverde de la Virgen exceeded 5 points on the 0–10 scale. In other words, these host cities are not in a very privileged position in terms of urban sustainability. This situation results in the thought and reflection that there is still much to be achieved for urban areas to contribute efficiently to the goals of the 2030 Agenda and the SDGs. In the case where the cities on The French Way of Saint James in Castilla y León were analyzed have to focus jointly on some interventions and action plans in particular, and the results have shown the need to act to increase resilience against the impacts of climate change and improve the sustainable management of resources. For the relationship between climate change and The French Way of Saint James in Castilla y León, climatic conditions with decreased rainfall and increased temperatures are projected. This has the potential to affect the seasonality of visits, the health of pilgrims, the conditions of public services in cities and towns, and even the cultural heritage [52]. Regarding resource management, a circular model of landscape resource regeneration can, therefore, contribute to the reconstruction of local economies. In small towns and in the innermost territories, where depopulation and human dynamics affect the quality of the landscape and elements of vitality persist, regenerative models of resources inspired by circular economic systems, such as slow tourism, can be applied with greater ease [53]. On the other hand, this set of cities shows its best performance in terms of sustainable mobility and intervention and governance instruments. Another important point that emerges from the joint analysis of the cities is the high degree of diversification and heterogeneity of the municipalities in relation to sustainability.

If the results are analyzed by differentiating two groups of cities, understanding that the present behaviors, resources, and realities of León, Burgos, and Ponferrada (larger cities) are different from Cacabelos, Astorga, and Valverde de la Virgen (smaller cities), especially due to the size of their populations, we find the following behaviors: larger cities have a better urban sustainability performance compared to small cities. This was evidenced in 8 of the 10 Goals (1, 2, 4, 5, 6, 7, 8, 9, 10). However, small cities, as a whole, present better performances in Objective 3 and 9 regarding climate change and innovation, respectively. This is mainly justified by the high values obtained by Valverde de la Virgen, since they are results, at least, that attract attention. Above all and especially, they draw attention to the issue of innovation since a great challenge and problem that the smaller towns and cities of Castilla y León have is their lack of connectivity and innovation.

The activity and the results obtained are useful since they can be part of a management tool that supports the prioritization of local and/or regional policy interventions based on reliable, verifiable, and up-to-date data. In addition, this article helps to improve the actions of The Way of Saint James and the relationships between cities and towns, as the Spanish Federation of Associations of Friends of The Way of Saint James demonstrates. This article stimulates the opportunity of local governments to implement local urban agendas, integrate urban policies, accelerate bottom-up actions, and establish innovative public policies, potentially linked by their synergy with The Way of Saint James. Its greatest

weakness is that the implementation of concrete actions depends on political will and even financial aid.

Considering the establishment of urban sustainability, the 2030 Agenda and the SDGs on The Way of Saint James as a development alternative can bring with them an opportunity to lead a paradigm shift in a group of cities (and also towns) that face great challenges at the local and regional level. The link, and the knowledge that they are working with the SDGs and the 2030 Agenda, generates the potential opportunity to obtain aid and subsidies (from interested companies, the European Union, or the Spanish government, for example) to execute action plans, with support based on the UAE, which is reliable for measuring progress and following up with updated data. The historical–traditional relationship that the analyzed cities from The French Way of Saint James present in this case in its passage through Castilla y León, in addition to delimiting the scope of the investigation, provides the possibility of strengthening existing alliances (Spanish Federation of Associations of Friends of The Way of Saint James) and promoting new ones. Although the definition and measurement of sustainability is a complex and debatable issue, the work places it at the center of the scene. What is not debatable is that cities must work more ambitiously in search of sustainable development and having a reliable diagnosis, as reflected in this article, is the central tool to direct interventions and monitor them with the passage of time (especially useful for smaller cities).

Future research could even use the method developed in this article with new data from the Spanish Urban Agenda, post-2022, so that the impacts of COVID-19 on the sustainable development of the analyzed cities are known. The monitoring of urban sustainability over time is a great tool for observing how cities are developing, how they are growing (or decreasing), and where the main weakness in their progress is. In addition, the SUA itself also makes it possible to compare the cities of The French Way of Saint James in Castilla y León with the cities of The French Way of Saint James in other Autonomous Communities such as Aragón, Navarra, La Rioja, or Galicia, for example, organizing the results by city (as in this case) or by Autonomous Communities.

The measures of change for a better sustainable performance of the cities analyzed must be based on the cultural-territorial union of the cities themselves with The Way of Saint James. In this way, by acting together, they can help each other by taking exemplary cases, transmitting knowledge and actions, and, above all, requesting help from various governmental or non-governmental organizations to apply them in a timely manner in specific action plans. Likewise, these cities must support and guide the towns on The Way of Saint James in Castilla and León, which depend largely on the passage of tourists, and which currently suffer from serious problems such as depopulation, aging, and lack of services, among others.

This research facilitates the debate on the progress and advancement of urban sustainability and the SDGs, stimulates research on the subject, promotes citizen awareness and participation from academic environments, and even engages local and regional authorities in the search for cities that are prepared for greater social welfare. For additional information on the subject see Supplementary Materials.

**Supplementary Materials:** The following supporting information can be downloaded at: https://www.mdpi.com/article/10.3390/su14159164/s1, Additional information on the subject can be found in the UVa Master's Thesis: "El camino hacia los ODS—Metodología para la localización de los objetivos de desarrollo sostenible en las ciudades y pueblos del Camino de Santiago" publicly available here. https://uvadoc.uva.es/handle/10324/45282?locale-attribute=es (accessed on 1 June 2022). The mentioned Master's Thesis has been recognized by the UVa with the Extraordinary End-of-Master Award.

**Author Contributions:** Conceptualization, F.T., L.F.L.-C. and O.F.B.; methodology, F.T. and M.S.W.; validation O.L.G.-N., D.A.M.V. and J.E.M.-V.; formal analysis, F.T., M.S.W. and O.L.G.-N.; investigation, F.T., L.F.L.-C. and M.S.W.; writing F.T. and O.F.B., writing—review and editing, F.T., L.F.L.-C., O.F.B. and M.S.W.; supervision, A.C.-G. All authors have read and agreed to the published version of the manuscript.

**Funding:** This research received no external funding.

**Institutional Review Board Statement:** Not applicable.

**Informed Consent Statement:** Not applicable.

**Data Availability Statement:** In this study, publicly available datasets from the Spanish Urban Agenda were analyzed. This data can be found here: https://www.aue.gob.es, accessed on 10 June 2021.

**Acknowledgments:** The main author of the article thanks his work team, his tutors and directors who motivate the writing of the article. The authors would like to thank the European Union for supporting this work through the FUSILLI project (H2020-FNR-2020-1/CE-FNR-07-2020). Francisco Tomatis has been financed under the call for UVa 2020 predoctoral contracts, co-financed by Banco Santander.

**Conflicts of Interest:** The authors declare no conflict of interest.

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
