# Peer review of "Evaluation of Urban Sustainability in Cities of The French Way of Saint James (Camino de Santiago Francés) in Castilla y León according to The Spanish Urban Agenda"

_sustainability, doi:10.3390/su14159164_

Round 1
Reviewer 1 Report
-The Introduction section and the section 2 The French Way of Saint James (“Camino de Santiago Francés”) in Castilla y León are a sort o mixture between several elements which are appropriate to an Introduction (i.e. what the study is about, its motivation, why the research topic/issues are important) and those which are useful for writing a section dedicated to the study-area. We suggest to clearly separate these two sections.
-The theoretical background research should be improved. in the current stage, the paper is focused only on SUA document which is specific to Spain.
-the research questions are nor clearly mentioned.
-lines 130-135 and lines 146-153 seem to be very similar and the Fig 2 which should be helpful for clarifying the differences between these two paragraphs is not visible. it is too small.
-the role of Fig. 1 is ...??? may be the authors should improve their comments and explanations for this issue. This Fig. 1 has no legend, scale, neighbourhood !
-Fig. 3 has no measure units
-the SDG 11 is mentioned for the first time in the article at line 75 but regarding the Table 2 - who know by heart all the SDGs by number? we suggest to restructured or to rethinking the Table 2.
-there is a reversal of table numbering: table no. 2 is before table no. 1 !
-in the current table 1 - it will be useful if a specification will be made concerning the specific objectives of SUA (meaning to be more clearly that they are the same mentioned in Table 2, in current numbering)
-lines 250-261 - please use the English versions of the official names . the Spanish names are nor useful the the readers.
-the current table 1 is to large ! maybe the font could be change .
-in Table 3 are 6 cities and in Fig. 4 are only 5 cities. Astorga is missing from the table.
Author Response
- The Introduction section and section 2 have been modified: The Introduction now refers to the importance of the research, the general context (urban sustainability, world population increase in cities, SDG, Spanish Urban Agenda) and briefly the methodology and research contribution. While section 2 covers the location and scope of the investigation, particularly characterizing The Way of Saint James, the cities analyzed and their current context.
- The main and major modifications of the article have focused on the theoretical background, opting to particularly explain concepts related to urban sustainability and its importance in measuring it. The global urban context and applications of SUA, among others, have also been expanded.
- - The research question is now present even within the Abstract, in such a way that it is clear and visible: where is The Fench Way of Saint James going respect to the urban sustainability of its host cities?
- - The lines that seemed to be similar (130-135 and 146-153) have now removed one of them. The previous Figure 2 has been removed and even replaced.
- The previous Figure 1 has been removed.
- The previous Figure 3 has been superseded. Now have units of measure.
- Table 2 has been supplemented with the current "Figure 5", so that each of the SDGs is visible.
- The numbering of the tables has been resolved.
- It has been mentioned that the SUA Objectives have been mentioned in the current Table 2 (in section 4.2 Data Normalization)
- The Spanish versions have been eliminated, since it is considered that they were not essential for understanding and even the English name of the different government portfolios can vary and be confusing to the official agencies of the Spanish state.
- The font of the current "Table 3" has been changed, so that it is easier to understand and does not take up so much space.
- Astorga has been added to the current "Table 4".
As you can see, the corrections have been applied according to the authors' criteria. The corrections are considered to have been very valuable and have greatly contributed to the quality of the article: THANK YOU!
Reviewer 2 Report
Comments and suggestions for the author:
Article describes SUA evaluation index of sustainable development potential, can really looking for countries in different parts of the city for the strategy of sustainable development, through the platform provided by the evaluation data and the related indicators, can see clearly in different cities in the main on the sustainable index of different performance, to realize the sustainable development potential in the visualization. The research content and relevant conclusions of this paper are indeed in line with the needs of countries and regions, and also provide a reference model for other countries after the epidemic.
There is a lack of research on the applicability of SUA at present in this paper, and the research conclusions presented at present are not able to obtain specific measures to shift to sustainable development. What we only see are the differences and deficiencies in the development of different cities.
Author Response
- Within the multiple corrections made, the SUA has been related to the European Agenda, further developing the importance of research in the context of the EU fundamentally. In addition, it has been detailed that future research can follow the line to compare cities from other autonomous communities of Spain related to The Way of Saint James, for which the research could be replicated in other regions.
- The theoretical framework has been expanded to demonstrate the importance of working and diagnosing sustainable cities. The importance and application of territorializing the SUA has been expanded, having provided greater bibliographies directly related to the SUA (current bibliography nº4,7 and 10 especially).
In the conclusions, weaknesses, capacities and opportunities presented by the research and diagnoses of the cities carried out have been incorporated. The conclusions have even been grouped according to the characteristics and nature of the cities.
As can be seen, several modifications have been made according to the suggestions, especially in the theoretical framework and in the conclusions. The authors consider the suggestions to be a great contribution, since with the changes made, the quality of the article has notably improved: THANK YOU.
Reviewer 3 Report
The article deals with a very current problem: the situation of the rural areas and the linked intermediate cities and the depopulation dynamics that occur in them. It presents the actions in favour of sustainability and the SDGs as a viable alternative to improve their environmental, social and economic areas, and undoubtedly it can be. Taking the Spanish Urban Agenda and the indicators method generated for it as a framework, the current situation of these cities can be evaluated and the assessment and monitoring of urban policies over time can be carried out. This is a tool that is currently being implemented at the national level. One of the challenges, as the article says, is to territorialize it so that it can be used in the definition of local strategies for urban transformation towards sustainability.
However, the work presented does not analyse the subject in sufficient depth, neither at a theoretical level nor in its application to the case study, and the structure and discourse of the article is not entirely clear. Some observations in this regard are detailed below:
- The context in which the research takes place is a rural area whose cities are in the process of depopulation. The introduction should detail more clearly the dynamics that occur in these circumstances, the adverse effect they can have on the region and why sustainability and specifically the Urban Agenda can be a driver of change in these dynamics.
- There is a reference to intermediate cities, and this term is even included in the title. But there is no description of what an intermediate city is. It would be necessary to look for the appropriate references and check whether the 6 municipalities analysed belong to this category.
- As stated in section 2, the Way of St. James is an identity element that gives a sense of belonging to the locals and acts as a link between towns, making them part of a single system. This characteristic is not included in the introduction nor is it taken into account in the research as a whole. The municipalities are analysed separately and compared with each other, but no overall reading is made in the results or in the reflections and proposals made in the final section of the discussion and conclusions.
- A sufficiently detailed analysis of the literature on the subject is not presented. The phenomenon of depopulation experienced by this type of urban systems has already been analysed in previous published studies. Some research has even studied how local actions linked to sustainability can promote change and revitalization of these areas. A search and analysis of these works should be carried out in order to establish the theoretical basis for the research presented here.
- The objective is not clearly expressed. In the introduction it is stated that "the aim is to make visible, diagnose and analyse some development patterns" and in section 3 it is mentioned that the synergies and contradictions of the cities evaluated in the achievement of sustainable development and the Specific Objectives of the SUA are to be identified. The criteria should be unified and, in addition, they should be in accordance with what is provided in the results and conclusions.
- Too much information is given about the Way of St. James, very general information, not very relevant to understand the analysis and the results presented below. However, the municipalities analysed are not well described, nor are their main characteristics or the existing links between them. For example, Figure 2 is not relevant and Figure 3 is neither very representative nor very detailed; the analysis should be carried out with population data from the 6 municipalities analysed. It is recommended to make a case study section where the most significant data for the research are given.
At the methodological and research proposal point of view, it is considered that the work presented here analyses in a superficial way the application of the SUA at the local level. Since the case study is so limited, the analysis should go further than the pure application of the indicators already calculated by the SUA. This could be a first approximation work, but would require a later approach to assess whether the results are validated or not and to identify, as stated in the objective, development patterns or possible synergies and contradictions. It is considered that this should constitute a phase of field work and contrast with reality, which has not been carried out in this work and which is indispensable for the contribution to be significant.
Author Response
- The article has incorporated the reality of depopulation in Castilla y León and its dynamics at a national and regional level. It has also raised capacities and opportunities presented by the cities analyzed by the SUA to advance towards their own sustainability and to jointly advance towards a commitment of the Camino de Santiago to the urban sustainability of their towns and cities.
- The concept of "intermediate cities" has been removed from the title. However, the definition of intermediate cities and the importance they have in the current context of Castilla y León has also been added to the text. Specific information is added for each analyzed city, in such a way that they can be differentiated between them (Leon, Burgos and Ponferrada on the one hand, and Cacabelos, Valverde de la Virden and Astorga on the other) according to their current number of inhabitants and their greater or lesser relationship with the concept of "intermediate cities".
- It has been added that The Way of Saint James is an element of identity that gives a sense of belonging to people and cities, in addition to acting as a link between towns, making them part of a single system territorial-cultural. Also, in the conclusions, the opportunities that cities have to work together for the urban sustainability of the host cities and towns of The Way of Saint James in Castilla y León have been added. The conclusions also comment on the potentialities of joint work, synergies and opportunities, based on the article, that cities have to ask for economic aid that can improve their urban sustainability, justified by the union with The Way of Saint James.
- Theoretical framework has been added regarding urban sustainability and the depopulation process of Castilla y León in particular. Bibliographies and reference articles have been incorporated in this regard.
- The criteria of the objectives have been unified: make visible, diagnose and analyze the urban sustainability of the cities of The Way of Saint James de Castilla y León. In this way, the results and conclusions respond to the main research objective, even identifying weaknesses, capacities and opportunities that cities present with this analysis.
- General information on The Way of Saint James has been removed. General descriptions of the analyzed cities have been added, differentiating their nature and the contexts of each one. Figure 2 has been eliminated and figure 3 has been replaced in such a way that the urban realities of each city are better visible.
- Although the work superficially analyzes each city with the SUA data, the research, through its publication, wishes to transcend the academic spheres to encompass reality. Although the investigation is methodological, it exposes the reality that even the local authorities themselves do not have such information and could use it for their personal benefit. The University of Valladolid plays an important role in the development of cities and towns in Castilla y León, a region that is going through a current crisis of depopulation and loss of agricultural activity. Precisely, the role of the University is to help change realities and link, in an academic way, with political and social decisions. The cities analyzed are known by the authors and have even been visited on more than one occasion for various reasons.
Round 2
Reviewer 3 Report
The effort that has been made in the revision of the article is appreciated but the contribution is not considered relevant enough to be published.